# Enhanced ε-Poly-L-Lysine Production in *Streptomyces albulus* through Multi-Omics-Guided Metabolic Engineering

**DOI:** 10.3390/biom14070752

**Published:** 2024-06-25

**Authors:** Liang Wang, Hao Yang, Mengping Wu, Hongjian Zhang, Jianhua Zhang, Xusheng Chen

**Affiliations:** Key Laboratory of Industrial Biotechnology, School of Biotechnology, Jiangnan University, Ministry of Education, Wuxi 214122, China; wangl@jiangnan.edu.cn (L.W.); 7220201025@stu.jiangnan.edu.cn (M.W.); jhzhang@jiangnan.edu.cn (J.Z.)

**Keywords:** ε-Poly-L-lysine, antimicrobial, preservatives, multi-omics, metabolic engineering

## Abstract

Safe and eco-friendly preservatives are crucial to preventing food spoilage and illnesses, as foodborne diseases caused by pathogens result in approximately 600 million cases of illness and 420,000 deaths annually. ε-Poly-L-lysine (ε-PL) is a novel food preservative widely used in many countries. However, its commercial application has been hindered by high costs and low production. In this study, ε-PL’s biosynthetic capacity was enhanced in *Streptomyces albulus* WG608 through metabolic engineering guided by multi-omics techniques. Based on transcriptome and metabolome data, differentially expressed genes (fold change >2 or <0.5; *p* < 0.05) and differentially expressed metabolites (fold change >1.2 or <0.8) were separately subjected to gene ontology (GO) and Kyoto Encyclopedia of Genes and Genomes (KEGG) pathway enrichment analysis. The integrative analysis of transcriptome, metabolome, and overexpression revealed the essential roles of isocitrate lyase, succinate dehydrogenase, flavoprotein subunit, diaminopimelate dehydrogenase, polyphosphate kinase, and polyP:AMP phosphotransferase in ε-PL biosynthesis. Subsequently, a strain with enhanced ATP supply, L-lysine supply, and ε-PL synthetase expression was constructed to improve its production. Finally, the resulting strain, *S. albulus* WME10, achieved an ε-PL production rate of 77.16 g/L in a 5 L bioreactor, which is the highest reported ε-PL production to date. These results suggest that the integrative analysis of the transcriptome and metabolome can facilitate the identification of key pathways and genetic elements affecting ε-PL synthesis, guiding further metabolic engineering and thus significantly enhancing ε-PL production. The method presented in this study could be applicable to other valuable natural antibacterial agents.

## 1. Introduction

ε-Poly-L-lysine (ε-PL) is a bioactive molecule composed of 25–35 L-lysine residues linked through α-carboxyl and ε-amino groups. It is known for its broad-spectrum antimicrobial activity, water solubility, biodegradability, and high safety. Studies have shown that ε-PL is non-toxic in the human body and can be fully metabolized [1]. Additionally, it is stable at high temperatures and does not form harmful byproducts [2]. These excellent properties have led to its widespread use as a food preservative in several countries. The significance of ε-PL as a food preservative lies in its ability to effectively combat the growth of foodborne pathogenic bacteria, which cause approximately 600 million illnesses and 420,000 deaths annually [3,4]. For example, ε-PL can effectively inhibit the growth of *Escherichia coli* O157 and *Listeria monocytogenes*, thereby extending the shelf life of raw milk at low temperatures from 8 days to 16 days [5]. Besides inhibiting the growth of various microorganisms, ε-PL also maintains the sensory quality and nutritional value of foods. For instance, adding ε-PL to fresh juices can significantly extend their shelf life and prevent microbial contamination [6]. In addition to its extensive applications in the food industry, ε-PL also shows great potential in the medical field. For example, ε-PL can be used to create hydrogels with self-healing and antibacterial properties, indicating its broad applications in wound repair and tissue engineering [7]. Research also indicates that ε-PL can bind to G-quadruplex DNA structures and effectively mediate targeted drug and gene delivery. These properties make ε-PL highly promising for anticancer drug delivery and gene therapy [8].

Historically, conventional mutagenesis has proven to be an effective approach to enhancing ε-PL production by screening mutants with high tolerance to the L-lysine analog (S-(2-aminoethyl)-L-cysteine) [9]. Moreover, using ribosome engineering to screen antibiotic (streptomycin, gentamycin, rifamycin, etc.)-resistant strains can also significantly increase ε-PL production [10]. However, these methods are time-consuming and labor-intensive, often requiring years of effort to develop a high-producing strain [11,12]. With the advancement of molecular biology, genetic recombination technology has been efficiently used to generate high-ε-PL producers. Hamano et al. employed site-directed mutagenesis to remove the feedback inhibition of aspartate kinase caused by L-lysine and L-threonine, resulting in a 33% improvement in ε-PL production [13]. Li et al. overexpressed the *dapA* gene in the L-lysine synthesis pathway to boost ε-PL biosynthesis in *Streptomyces diastatochromogenes* [12]. Various studies have indicated that enhancing the expression of ε-PL synthetase [14], increasing energy supply [15], ensuring nitrogen availability [16], and enhancing acid resistance [17] can all contribute to efficient ε-PL biosynthesis. However, owing to the complexity of cell metabolism, the inherited genetic background of *S. albulus* is only partly understood, and the unidentified rate-limiting steps in ε-PL metabolism restrict further research on engineering high-producing ε-PL strains.

Recently, omics technologies, including genomes, transcriptomes, proteomes, and metabolomes, have emerged as powerful tools for elucidating the genetic basis of superior phenotypes. When integrated with accurate genome modification techniques, omics technologies can guide the metabolic engineering of strains. For instance, to identify the key nodes for riboflavin metabolism regulation caused by *Bacillus subtilis* under oxygen-limited environments, the transcriptomes were compared and analyzed at different dissolved oxygen levels. It was found that dissolved oxygen regulates riboflavin biosynthesis by influencing the nitrogen metabolism regulators *glnR* and *tnrA*, as well as the purine metabolism inhibitor *purR*. Deleting these genes led to a significant increase in riboflavin production, with a concentration of 10.71 g/L, representing a 45.51% enhancement compared with the original strain [18]. Similarly, the results of genome and transcriptome analysis have suggested that sRNA NC-1 could be a crucial regulatory factor in affecting the acid tolerance and nisin production of *Lactococcus lactis* ssp. *Lactis*. Overexpression of sRNA NC-1 has led to a significant enhancement in both the acid tolerance and nisin production of *L. lactis* F44, with 2.23-fold and 2.33-fold increases, respectively [19].

In our previous study, a high-producing strain, *S. albulus* WG608, was obtained through successive rounds of mutagenesis and ribosome engineering [20]. This study attempts to further enhance the ε-PL synthetic capacity of *S. albulus* WG608 through metabolic engineering. However, owing to the complexity of cellular metabolism, the efficient synthesis mechanism of ε-PL in WG608 is not yet clear, which limits the applications of metabolic engineering. Therefore, multi-omics analysis was used to guide further metabolic engineering and improve ε-PL production. First, transcriptional and metabolic changes between high- and low-producing strains were compared and evaluated at 48 h, 96 h, and 144 h to reveal differences in ε-PL synthesis pathways. Subsequently, the key pathways and gene elements affecting ε-PL synthesis were identified through overexpression, and ε-PL production was further improved via metabolic engineering. Finally, the fermentation performance of the resulting strain (WME10) was evaluated in a 5 L bioreactor. This work provides valuable genetic resources for studying the mechanism of ε-PL synthesis and a theoretical basis for constructing an efficient ε-PL synthesis cell factory through metabolic engineering.

## 2. Materials and Methods

### 2.1. Strains, Plasmids, and Media

*S. albulus* WG608 is a high-producing mutant derived from the original strain, *S. albulus* M-Z18, through multiple rounds of mutagenesis combined with ribosome engineering [20]. *S. albulus* WG608 was deposited at the China Center for Type Culture Collection (CCTCC M2019589) and used as an original strain in this study. *Escherichia coli* DH5α was used as cloning and expression hosts, while *E. coli* ET12567/pUZ8002 was employed for *Streptomyces*–*E. coli* interspecies conjugation to introduce plasmids into *Streptomyces* [21].

*S. albulus* strain spores were cultured at 30 °C on BTN medium (10 g/L glucose; 2 g/L fish meal peptone; 1 g/L yeast powder; 20 g/L agar powder), with the pH adjusted to 7.5 using a NaOH solution [10]. *E. coli* strains were cultured at 37 °C in LB medium containing 10 g/L tryptone, 5 g/L yeast powder, and 10 g/L NaCl [21]. The pH was adjusted to 7.5 using the NaOH solution, with final concentrations of 25 μg/mL kanamycin, 50 μg/mL apramycin, and 25 μg/mL chloramphenicol added, if necessary. For intergeneric conjugation between *S. albulus* and *E. coli* at 30 °C, MS medium (20 g/L mannitol; 20 g/L soybean powder; 10 mM MgCl_2_; 20 g/L agar powder; pH 7.0) was utilized [17]. Recombinant colonies were screened on an MS medium with final concentrations of 50 μg/mL apramycin and 25 μg/mL nalidixic acid after 16–18 h of cultivation. *S. albulus* shake-flask fermentation was carried out in a YP medium, and *S. albulus* fed-batch fermentation was carried out in a YHP medium, as described by Wang et al. [10].

### 2.2. Plasmid and Strain Construction

The strains and plasmids used in this study are listed in Table 1, while the primers are detailed in Appendix A. The overexpression experiments in WG608 were conducted following the protocol described by Zhang et al., with slight adjustments [17]. To overexpress *zwf* (encoding glucose-6-phosphate dehydrogenase) from WG608, the DNA fragment containing the *zwf* coding region was amplified from the genomic DNA of WG608 using the primer pair *zwf*-F/-R. Subsequently, the purified fragment was ligated into the Nde I/EcoR I-digested vector pIB139 [22]. The resulting ligation products were then transformed into competent *E. coli* DH5α, and exconjugants were selected from LB plates containing 50 μg/mL apramycin. Following validation through colony PCR with corresponding primers and DNA sequencing (Aenta, Suzhou, China), the overexpression vector pIB139-*zwf* was obtained. The DNA sequences encoding *vgb* (encoding *Vitreoscilla* hemoglobin), *ddh^bs^*, and *ddh^cg^* (encoding diaminopimelic dehydrogenase from *Bacillus sphaericus* and *Corynebacterium glutamicum*, respectively) were chemically synthesized (Aenta, Suzhou, China) with codon optimization. The remaining genes used in this study were amplified as described previously.

The co-expression of *ppk2B^cg^* (encoding polyphosphate kinase from *Corynebacterium glutamicum*) and *pap* (encoding polyP:AMP phosphotransferase from *Acinetobacter johnsonii*) was conducted following the method outlined by Lv et al., with some adaptations [23]. The DNA fragment containing *ppk2B^cg^* was amplified from pIB139-*ppk2B^cg^* using the primer pair c-*ppk2*-F/R, while the DNA fragment containing *pap* was amplified from pIB139-*pap* using the primer pair c-*pap*-F-2/-R-2. Subsequently, the co-expression vector pIB139-*ppk2B^cg^-pap* was constructed according to the aforementioned procedure.

For the co-expression of *aceA*, *sdhA*, *ddh^cg^*, *pap*, *ppk2B^cg^*, and *pls* in WG608, the *pls* fragments were first amplified from the genome of WG608 using the appropriate primers. Subsequently, the obtained DNA fragments were ligated into the *EcoR* I-digested vector pIB139-*ppk2B^cg^*-*pap*. Additionally, the *aceA*, *sdhA*, and *ddh^cg^* fragments were amplified from pIB139-*aceA*, pIB139-*sdhA*, and pIB139-*ddh^cg^*, respectively. These DNA fragments were then ligated into the *Nde* I/*Eco*R I-digested vector pIB139. The ASD (*aceA-sdhA-ddh^cg^*) fragments were amplified from pIB139-*aceA-sdhA-ddh^cg^*. The resulting ASD fragment and *PermE** were subsequently ligated into pIB139-*ppk2B^cg^-pap-pls* to generate the co-expression vector pIB139-*ppk2B^cg^-pap-pls-aceA-sdhA-ddh^cg^*.

Finally, the recombinant plasmids mentioned above were individually transformed into *E. coli* ET12567 for intergeneric conjugation with WG608 [24]. To identify overexpression strains, the transformants were screened on solid BTN medium supplemented with apramycin and nalidixic acid, and the colonies were confirmed via PCR using the relevant primers.

### 2.3. Shake-Flask, Batch, and Fed-Batch Fermentation of S. albulus

For shake-flask fermentation, WG608 and its mutants were cultured in a YP medium at 30 °C and 200 rpm for 24 h to generate a seed culture. Subsequently, the seed solution was then inoculated with an 8% volume fraction and further incubated in a YP medium for 72 h at 30 °C and 200 rpm [10].

Batch fermentations of WG608 and its mutants were carried out in 1 L bioreactors [17]. The initial pH was adjusted to 6.80 by adding ammonia solution (12.5%, *v*/*v*), followed by inoculating 60 mL (8% inoculation volume) of seed solution into the YHP medium. The pH of the fermentation broth was regulated at approximately 4.0 by automatically adding ammonia solution. The dissolved oxygen (DO) level was maintained above 20% air saturation, controlled by adjusting the agitation speed from 200 to 1000 rpm.

Fed-batch fermentations of WG608 and its mutants were carried out in 5 L bioreactors (Biotech-5BG, Baoxing Bio-Engineering Equipment Co., Ltd., Shanghai, China). The initial pH was adjusted to 6.80 by adding ammonia solution (12.5%, *v*/*v*), followed by inoculating 60 mL (8% inoculation volume) of seed solution into the YHP medium. The polyP_6_ addition strategy was consistent with the description by Yang et al. [15]. The pH of the fermentation broth was regulated at approximately 4.0 by automatically adding ammonia solution. The dissolved oxygen (DO) level was maintained above 20% air saturation, controlled by adjusting the agitation speed from 200 to 1000 rpm. Moreover, when the glucose concentration in the fermentation broth was below 10 g/L, a sterilized glucose solution (80%, *w*/*v*) was automatically added and maintained at around 10 g/L. When the ammonia nitrogen (NH_4_^+^-N) concentration decreased below 0.5 g/L, a sterilized (NH_4_)_2_SO_4_ solution (40%, *w*/*v*) was automatically added and maintained at around 0.5 g/L to include a nitrogen source.

### 2.4. Analytical Methods

The detection of fermentation parameters was carried out as described by Wang et al. [10]. A 10 mL sample of fermentation broth was centrifuged at 17,843× *g* for 10 min to isolate the mycelia. The mycelia were filtered through pre-dried and pre-weighed filter paper and then dried at 105 °C until a constant weight was achieved to determine the biomass. The supernatant was used to assess the ε-PL concentration following the procedure described by Itzhaki [25]. Glucose concentration was measured using a biosensor (SBA-40B, Shandong Academy of Sciences, Jinan, China). The NH_4_^+^-N concentration was determined via Nessler reagent spectrophotometry [20]. The intracellular ATP levels were quantified with the ATP Assay Kit (Beyotime, Suzhou, China) following the manufacturer’s instructions.

### 2.5. RNA Sequencing and Transcriptome Analyses

Transcriptome analysis samples were collected at 48 h, 96 h, and 144 h intervals. RNA extraction was performed following the Total RNA Extractor protocol (Trizol) (Sangon, Shanghai, China). The isolated RNA was reverse-transcribed after digesting residual gDNA using HiScript^®^ III RT SuperMix for qPCR (+gDNA wiper) (Vazyme, Nanjing, China). To minimize sequencing interference, DNase I (NEB, Beverly, MA, USA) was used for DNA removal, while rRNA was digested using the RiboCop rRNA Depletion Kit for Gram-Positive Bacteria (G^+^). The resulting RNA underwent fragmentation with RNA Fragmentation Buffer and was reverse-transcribed using an N6 randomized primer to produce double-stranded cDNA. The 5′ ends of the cDNA fragments were phosphorylated, and A bases were added to the 3′ ends for the DNA fragments’ ends. Following ligation, PCR amplification was carried out using specific primers. The resulting PCR product was denatured using heat to obtain single-stranded DNA, which was then used to construct a single-stranded circular DNA library through circularization [26].

The DNA library was sequenced using the Illumina HiSeqTM 2000 (Illumina, CA, USA) to assess if it was suitable for future analysis. Raw data were processed to exclude statistical comparison rates and read distributions based on the reference sequence. The refined data were then aligned to the genome of *S. albulus* ZPM (NCBI accession number NZ_CP006871) using SOAPaligner/SOAP2. Genes meeting the criteria of FDR < 0.001, *p*-value < 0.05, and |log_2_FoldChange| ≥ 1 were identified as differentially expressed genes (DEGs). Subsequently, the DEGs were subjected to GO enrichment analysis (http://www.geneontology.org/, accessed on 12 June 2024) and KEGG pathway enrichment analysis (http://www.genome.jp/kegg/pathway.html, accessed on 12 June 2024) [27].

### 2.6. Metabolomics Analysis

Samples were collected from a 5 L fermenter at 48 h, 96 h, and 144 h. In total, 5 mL of each sample was divided into two equal parts: one for liquid chromatography analysis, and the other for the lyophilized dry weight measurement. To detect metabolites, an equal volume of methanol was added to 2 mL of fermentation broth, mixed at 4 °C for 30 min, and centrifuged at 17,843× *g* for 15 min to obtain the supernatant, which was dried under vacuum. The samples were re-dissolved in 500 µL of 50%, 75%, and 90% methanol in water. The supernatant obtained via centrifugation at 20,817× *g* for 15 min was then used for UPLC-TOF-MS analysis. Each sample was analyzed three times in a single day for intra-day repeatability evaluation and over three different days for inter-day precision evaluation [28].

The metabolic profiles of *S. albulus* were analyzed via ultra-performance liquid chromatography–mass spectrometry (UPLC-MS). Chromatographic separation was carried out using the Thermo ScientificTM DionexTMUltiMateTM 3000 Rapid Separation LC system (ThermoFisher, Beverly, MA, USA), equipped with a reversed-phase C18 column (ACQUITY UPLS CSH column, 1.7 µm, 2.1 × 100 mm; Waters, USA) and a hydrophilic interaction liquid chromatography (HILIC) column (BEH Amide column, 1.7 µm, 2.1 × 100 mm; Waters, USA). The chromatographic column was used with the following liquid chromatography conditions: column temperature was 45 °C, the constant flow rate was 0.4 mL/min, and the injection volume was 2 µL. Samples were rapidly eluted using 0.1% formic acid in water (solvent A) and 0.1% formic acid in acetonitrile (solvent B). The separation was achieved with gradients of 95:5 *v*/*v* (solvent A/solvent B) at 0 min, 10:90 *v*/*v* at 10.0 min, 10:90 *v*/*v* at 11.0 min, 95:5 *v*/*v* at 11.1 min, and 95:5 *v*/*v* at 14.0 min.

The UPLC-MS analysis was performed using a Q-TOF Premier mass spectrometer operating in electrospray ionization mode (Waters Corp., Milford, MA, USA) with the following settings: positive ion mode; capillary voltage at 3.0 kV; cone voltage at 35.0 V; source temperature at 100 °C; desolvation temperature at 35 °C; desolvation gas flow at 600.0 L/h; collision energy at 6.0 eV; scan range *m*/*z* at 80–2000; scan time at 0.3 s; inter-scan time at 0.02 s. The TOF analyzer was in V mode with a high-resolution setting of 9000. Sodium formate was employed for calibration, and leucine–enkephalinamide acetate (LEA) was used as the lock mass (*m*/*z* 556.2931 in ESI+) at a concentration of 200 ng/mL and a flow rate of 0.1 mL/min.

Data preprocessing was performed with MassLynx applications manager version 4.1 (Waters, MA, USA). For untargeted analysis, MarkerLynx (v4.1, Waters, MA, USA) was used to integrate and align MS data points and convert them into exact mass retention time pairs. Principal component analysis (PCA), partial least squares–discriminant analysis (PLS-DA), and orthogonal partial least squares–discriminant analysis (OPLS-DA) were performed using Pareto scaling on all features with MarkerLynx) [12].

### 2.7. Statistical Analysis

All experiments were performed in triplicate, and all data are expressed as mean ± standard deviation. Statistical analysis was performed using SPSS (version 22.0, SPSS Inc., Chicago, IL, USA) with one-way analysis of variance (ANOVA) and Tukey’s test at *p* < 0.05.

## 3. Results and Discussion

### 3.1. Fermentation Performance of ε-PL High- and Low-Producing Strains

*S. albulus* WG608 is an ε-PL high-producing mutant derived from the original strain, *S. albulus* M-Z18, through rounds of mutagenesis, genome shuffling, and ribosome engineering [20]. To evaluate the difference in fermentation performance between WG608 and M-Z18, these two strains were separately subjected to fed-batch fermentation in 5 L fermenters. When the pH of the fermentation broth gradually decreased from the initial value of 6.8 to around 4.0, ammonia was continuously added to maintain it at 4.0. To achieve higher production, the glucose solution and (NH_4_)_2_SO_4_ solution were continuously added when they were nearly depleted, keeping their concentrations controlled at approximately 10 g/L and 0.5 g/L, respectively, throughout the fed-batch process. After 192 h of culturing, the ε-PL production and average specific ε-PL formation rate of WG608 were 44.33 g/L and 0.24 d^−1^, 23.9% and 83.38% higher than M-Z18 (Appendix A). It is difficult to further increase ε-PL production using the conventional mutagenesis method.

Moreover, owing to the long biosynthesis pathway and complex regulatory mechanisms involved in the de novo synthesis of ε-PL from glucose, identifying key genes affecting ε-PL biosynthesis in *S. albulus* is challenging. Transcriptome and metabolome analysis can reveal changes in cell metabolism by assessing global gene transcription and metabolite changes, respectively [29,30,31]. Therefore, a comparative study of the transcriptomes and metabolomes of M-Z18 (low-producing ε-PL strain) and WG608 (high-producing ε-PL strain) was conducted at the mid-log phase (48 h), late-log phase (96 h), and stationary phase (144 h) of fermentation (Appendix A) to identify the key metabolism pathways and elements affecting ε-PL biosynthesis.

### 3.2. Global Gene Expression and Metabolite Changes at Different Time Points

A total of 3632, 3968, and 3636 differentially expressed genes (DEGs) (fold change of >2 or <0.5, *p* < 0.01) were identified at 48 h, 96 h, and 144 h, respectively (Figure 1A). Pearson correlation analysis revealed significantly distinct expression patterns between WG608 and M-Z18 at each time point (PCC < 0.5) (Figure 1B). Compared with the control, the number of downregulated DEGs in WG608 exceeded that of upregulated DEGs. In addition, 2493 genes showed significant changes during the whole fermentation process, including 982 upregulated DEGs and 1491 downregulated DEGs (Figure 1C). These DEGs were further classified according to KEGG annotations to display their KEGG distribution characteristics. Both the upregulated and the downregulated DEGs were predominantly involved in carbohydrate metabolism and amino acid metabolism (Appendix A).

For the same strain, metabolome analysis showed that there were small differences between the mid-log phase and late-log phase and substantial differences between the mid-log phase and stationary phase. The discrete distribution between the M-Z18 and WG608 strain samples indicated that the metabolic levels of these two strains were significantly different at each time point (Figure 2A,B). Using the partial least squares–discriminant analysis model and univariate analysis (fold change of >1.2 or <0.83, Q < 0.05), 49, 57, and 64 differentially expressed metabolites (DEMs) were identified at 48 h, 96 h, and 144 h, respectively (Figure 2C). Consistent with the transcriptomic results, the majority of DEMs were primarily associated with carbohydrate metabolism and amino acid metabolism (Appendix A).

After comprehensively analyzing the transcriptomes and metabolomes, it was found that the DEGs and DEMs at 48 h, 96 h, and 144 h were mainly associated with carbohydrate metabolism and amino acid metabolism. These DEGs and DEMs were further analyzed to identify the key genes affecting ε-PL biosynthesis.

### 3.3. Transcriptional and Metabolic Changes Involved in ε-PL Biosynthesis at Different Time Points

#### 3.3.1. Upregulation of Glycolysis Pathway and Glyoxylate Cycle Enhanced the Synthesis of L-lysine Precursor Oxaloacetate

In the present study, the de novo synthesis pathway of ε-PL was constructed based on *Streptomyces albulus* ZPM genomic data (GenBank: CP006871.1); transcriptomic and metabolomic variations between the high-producing strain WG608 and the control are shown in Figure 3. ε-PL is polymerized from the precursor L-lysine through a membrane-bounded non-ribosomal synthetase ε-PL synthetase (Pls) via adenylation and sulfhydrylation activation, which means that L-lysine, Pls, and ATP play crucial roles in ε-PL biosynthesis. In *S. albulus*, the carbon skeleton of L-lysine is mainly derived from carbohydrate metabolism [32,33,34,35]. Therefore, transcriptional and metabolic changes in the carbohydrate metabolism pathways—including the glycolysis pathway, the pentose phosphate pathway (PP pathway), and the tricarboxylic acid cycle (TCA cycle)—were first analyzed to identify the key pathways for enhancing L-lysine supply.

In the glycolysis pathway, the *glk* gene (encoding glucokinase) was upregulated at 48 h, 96 h, and 144 h by 6.76-, 4.27-, and 2.79-fold, respectively, facilitating the conversion of glucose into glyceraldehyde 3-phosphate. Additionally, *pfk* (encoding phosphofructokinase) and *fbaA* (encoding triosephosphate isomerase) were upregulated at 96 h and 144 h, and *tpiA* was upregulated at 48 h (Table 2). Upregulating these genes led to increased levels of glyceraldehyde 3-phosphate and phosphoenolpyruvate at 48 h, 96 h, and 144 h (Appendix A), indicating the significant enhancement of the glycolysis pathway in WG608 throughout the whole fermentation process. In addition, the transcription level of pyruvate orthophosphate dikinase (encoded by *ppdk*) was also significantly upregulated at 48 h, 96 h, and 144 h, suggesting that the pyruvate supply was enhanced in WG608. However, contrary to expectations, the pyruvate was downregulated by 2.73- and 3.54-fold at 48 h and 96 h, respectively (Appendix A; Figure 3). These results indicate that more pyruvate was consumed in WG608, providing more intermediate metabolites for L-lysine biosynthesis in WG608.

The PP pathway is another route of oxidative decomposition for glucose and mainly provides NADPH and pentose for *S. albulus.* In the PP pathway, the transcription level of *pgl* (encoding phosphogluconolactonase) was upregulated by 2.42- and 2.19-fold at 48 h and 96 h, respectively. *tktA* (encoding transketolase) was upregulated at 48 h but significantly downregulated at 96 h. Furthermore, *gnk* (encoding gluconokinase) was significantly upregulated by 7.40-, 6.20-, and 3.10-fold at 48 h, 96 h, and 144 h, respectively. Upregulating *pgl* and *gnk* supplemented the high-producing strain with more gluconate-6-phosphate, facilitating the production of more ribulose-5P catalyzed by 6-phosphogluconate dehydrogenase. This also explains the significant upregulation of ribulose-5P at 48 h, 96 h, and 144 h (Figure 3; Appendix A). Sedroheptulose-7P was downregulated at 48 h and upregulated at 96 h and 144 h, which was consistent with the gene expression pattern of *tktA*. Overall, the PP pathway exhibited significant upregulation in the early fermentation phase, supplying more NADPH cofactors and precursors for cell growth and L-lysine biosynthesis. In the mid and late phases of fermentation, more L-lysine was generated and consumed for ε-PL synthesis. However, the PP pathway was downregulated, suggesting an inadequate supply of intracellular NADPH in the high-producing strain. This inference was confirmed by the decreased intracellular NADPH concentration in WG608 (Figure 4A).

The TCA cycle represents the primary hub for carbon skeletons that are essential for energy production and metabolism, providing energy and precursors for carbohydrate, lipid, and amino acid biosynthesis [36]. Gene expressions of *lcd*, *korAB*, and *sucCD* in the TCA cycle were significantly downregulated at 144 h (Figure 3; Table 2), and *pdhD* was upregulated at 48 h and 96 h. These findings suggest a weakened TCA cycle, particularly from the isocitrate to succinate stage, in the high-producing strain (Figure 3). Downregulating the TCA cycle led to a decreased NADH supply, which may limit ATP synthesis (Appendix A). Additionally, citric acid was upregulated at 144 h, while succinate was significantly downregulated at 48 h, 96 h, and 144 h, suggesting that the TCA cycle’s intermediate metabolites were far more depleted in the mid and late phases of fermentation. The glyoxylate cycle serves as a shunt or short alternative to the TCA cycle, in which the decarboxylation steps are bypassed by reactions catalyzed by isocitrate lyase (encoded by *aceA*) and malate synthase (encoded by *aceB*). *aceB* was significantly upregulated at 48 h and 96 h; *sdhA* was also upregulated at 48 h, 96 h, and 144 h. It is reasonable to hypothesize that the enhanced glyoxylate cycle improves the efficiency of four-carbon compound synthesis from acetyl-CoA, providing more oxaloacetate for L-lysine synthesis at the same time.

Additionally, downregulating the PP pathway may reduce the NADPH supply, while enhancing the NADPH supply in the high-producing strain may benefit L-lysine and ε-PL biosynthesis in WG608. For example, Liu et al. promoted L-lysine biosynthesis by reconstructing the diaminopimelic acid pathway and overexpressing enzymes related to NADPH generation and utilization, such as glucose-6-phosphate dehydrogenase and 6-phosphogluconate dehydrogenase [37]. Liu et al. employed systems metabolic engineering methods to enhance NADPH self-regulation and ATP supply in *Corynebacterium glutamicum*, thereby improving the intracellular metabolic state and increasing L-lysine production [38]. In summary, strengthening the glyoxylate cycle and NADPH supply may benefit L-lysine and ε-PL biosynthesis in WG608.

#### 3.3.2. Enhanced Biosynthesis of L-Lysine

L-lysine is the only precursor of ε-PL biosynthesis and is biosynthesized by the diaminopimelic acid pathway in *S. albulus*. Figure 3 shows that *asd* (encoding aspartate–semialdehyde dehydrogenase) was significantly upregulated at 48 h and 96 h, while *lysA* (encoding diaminopimelate decarboxylase) and *dapA* (encoding 4-hydroxy-tetrahydrodipicolinate synthase) were downregulated at 96 h and 144 h, indicating that the diaminopimelic acid pathway was upregulated at 48 h but downregulated in the late phase of fermentation. Additionally, *racD*, *asnB*, *metH*, *thrC*, and *metE*—involved in the D-aspartate, L-asparagine, L-threonine, and L-methionine biosynthesis pathways, respectively—were all significantly downregulated at 48 h and 96 h (Appendix A). Some competitive pathways of the L-lysine biosynthesis pathway were also repressed during the whole fermentation process, including L-threonine, L-methionine, D-aspartate, and L-asparagine. These results suggest that the L-lysine biosynthesis pathway was enhanced and the biosynthesis of many other amino acids was weakened in WG608, which contributes to WG608’s function as a highly efficient microbial cell factory for L-lysine. These results also explain the upregulated L-lysine in WG608 during the whole fermentation process (Appendix A). Some studies have shown that weakening the competitive amino acid synthesis pathways can significantly enhance L-lysine production. For instance, Shu et al. used metabolic engineering techniques to block the L-ornithine and L-arginine synthesis pathways, thereby accumulating more intermediates for L-lysine synthesis and significantly improving L-lysine production efficiency [39]. This result is consistent with our findings. Additionally, research indicates that the expression of the *dapA* and *lysA* genes is constitutive during cell growth and is not regulated by feedback inhibition from diaminopimelate or lysine. Therefore, it can be inferred that the downregulation of *dapA* and *lysA* genes in WG608 during the late fermentation stage is not due to the accumulation of excess L-lysine [40]. Given that the expression levels of *lysA* and *dapA* are significantly lower in WG608 compared with the original strain at the late fermentation phase (144 h), introducing new L-lysine biosynthesis pathways may be an effective strategy to further enhance L-lysine and ε-PL biosynthesis.

#### 3.3.3. Enhanced ε-PL Biosynthesis

The ε-PL synthase (Pls) is a crucial rate-limiting enzyme in ε-PL synthesis. It is a non-ribosomal peptide synthetase (NRPS) consisting of six transmembrane domains (TM1-TM6) and three tandem domains (C1-C3). During ε-PL biosynthesis, the carboxyl group of the L-lysine monomer is adenylated and then transferred to the thiol group of the ε-PL synthase active site, generating an activated aminoacyl thioester intermediate [13]. PLs catalyze the formation of repetitive peptide bonds between sulfurized lysine and the ε-amino group of free L-lysine or ε-PL peptides, gradually elongating ε-PL. These detailed processes were described by Hamano [41]. Table 2 shows the expression levels of Pls in the high-producing strain were 3.71, 5.54, and 2.19 times higher than those in the original strain at 48, 96, and 144 h, respectively. This indicates that WG608 maintains high Pls expression throughout the fermentation process, which is closely related to increased ε-PL production.

### 3.4. Improvement in ε-PL Production via Metabolic Engineering

#### 3.4.1. Increasing ATP Supply and Pls Expression for ε-PL Production

ATP plays a vital role in ε-PL biosynthesis. In *S. albulus*, ATP is primarily synthesized through the glycolysis pathway and the electron transport chain. Transcriptomic analysis reveals significant glycolysis pathway upregulation but notable TCA cycle downregulation during fermentation, indicating a potential decrease in NADH supply that could further affect ATP generation. Appendix A show that the intracellular ATP and NADH levels of WG608 were higher than those of the original strain at the early phase of fermentation (48 h) but significantly lower at the mid and late phases of fermentation (96, 144, and 192 h). These findings suggest that the extensive synthesis of ε-PL during the late fermentation phase leads to an inadequate supply of ATP in WG608.

Here, two methods were chosen to enhance the ATP supply in WG608 (Figure 4): (1) the expression of *Vitreoscilla* hemoglobin (VHb, encoded by the *vgb* gene); (2) the construction of an ATP regeneration system. VHb, a bacterial hemoglobin, binds and delivers oxygen to respiratory oxidase to enhance oxidative phosphorylation and promote ATP generation [33]. In addition, based on the types of metal ions in the fermentation broth, polyP:AMP phosphotransferase (encoded by *pap*) from *Acinetobacter johnsonii* was chosen to catalyze the conversion of AMP into ADP [42,43], while polyphosphate kinase (encoded by *ppk2B^cg^*) from *Corynebacterium glutamicum* was selected to catalyze the conversion of ADP into ATP [44,45,46]. Consequently, the following strains were constructed (Figure 4A,B): recombinant strain WME1 for overexpressing *vgb* in WG608; recombinant strain WME2 for co-expressing *pap* and *ppk2B^cg^* in WG608; and recombinant strain WME3 for overexpressing ε-PL synthetase (encoded by *pls*) in WG608. As a result, the ε-PL production rates of WME1 and WME2 were 2.21 ± 0.06 g/L (*p* ≤ 0.01) and 2.34 ± 0.09 g/L (*p* ≤ 0.001), which were 14.50% and 21.24% higher than the ε-PL production of WG608, respectively (Figure 4C). This indicates that the co-expression of *pap* and *ppk2B^cg^* is more effective in enhancing WG608 production. Hence, the overexpression of *ppk2B^cg^* can be deemed more suitable for increasing the ATP supply in WG608 compared with the overexpression of *vgb*.

**Figure 4 biomolecules-14-00752-f004:**
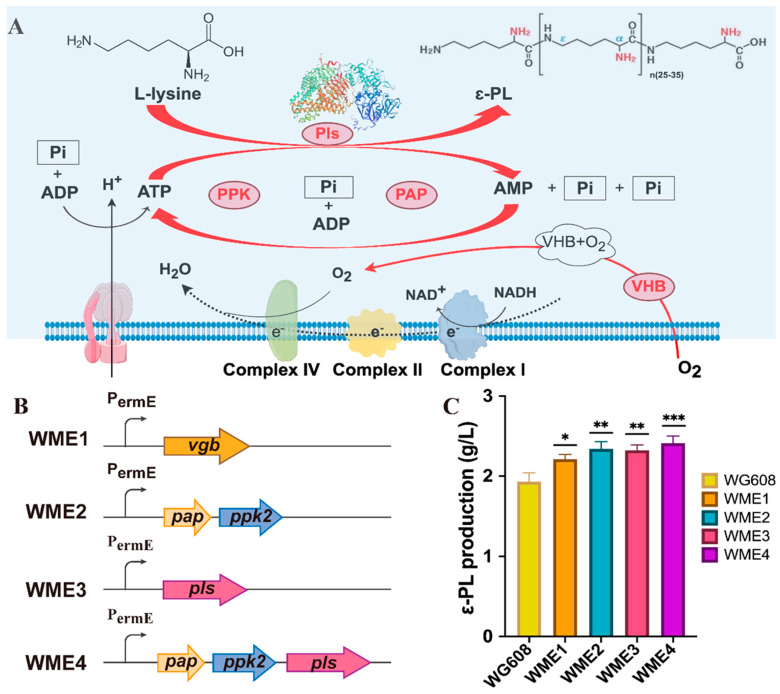
Effect of the increasing ATP supply and overexpressing ε-PL synthetase on ε-PL production. (**A**) Engineering for enhancing ε-PL synthesis in *S. albulus* WG608. (**B**) Schematic of the construction of strains WME1-WME4. (**C**) Effect of the overexpressing different modules on ε-PL production. *pls*, *ppk*, *pap,* and *vgb* represent the genes encoding ε-PL synthetase, polyphosphate kinase, polyP: AMP phosphotransferase and *Vitreoscilla* hemoglobin, respectively. All data were from biological triplicates. Error bars represent standard deviation. The statistical analysis was performed by one-way ANOVA analysis; *, **, and *** indicate *p* ≤ 0.05, *p* ≤ 0.01, and *p* ≤ 0.001 relative to the control (WG608), respectively.

Figure 4B,C show that PLs overexpression (WME3) also led to a 20.20% improvement in ε-PL production. Subsequently, a recombinant strain, WME4, was constructed with *ppk2B^cg^*, *pap*, and *pls* gene co-overexpression. The ε-PL production of WME4 reached 2.41 ± 0.09 g/L in a shake flask, showing no significant difference compared with that of WME2 and WME3. These results indicate that increasing ATP supply and Pls expression can effectively improve ε-PL production. When simultaneously enhancing the supply of ATP and Pls expression, the intracellular L-lysine supply in WG608 may be insufficient, thereby limiting further increases in ε-PL production.

#### 3.4.2. Increasing L-lysine Supply for ε-PL Production

Based on our integrative analysis of transcriptomes and metabolomes at 48 h, 96 h, and 144 h, several hypotheses on ways to increase the L-lysine supply in WG608 can be propounded: (1) increase the NADPH supply; (2) enhance the glyoxylate pathway; and (3) introduce a new L-lysine biosynthesis pathway.

NADPH is a crucial cofactor in the L-lysine biosynthesis pathway, as the synthesis of 1 mol of L-lysine requires 4 mol of NADPH [47]. Figure 5A,B show that both the NADPH levels and NADPH/NAPD^+^ ratios in WG608 were significantly lower compared with those in the control throughout the fermentation process. This indicates that more NADPH was used in L-lysine biosynthesis in WG608, leading to the generation of more NADP^+^. *zwf* (encoding glucose-6-phosphate dehydrogenase) was overexpressed in WG608 to construct strain WME5 (Figure 5C). However, there were no discernible differences in ε-PL production between WME5 and WG608 (Figure 5D).

To increase the oxaloacetate supply for L-lysine biosynthesis, strain WME6 and strain WME7 were constructed by overexpressing *aceA* and *sdhA*, respectively, in WG608 (Figure 5C). The ε-PL production of WME6 and WME7 reached 2.33 ± 0.18 and 2.26±0.05 g/L, representing increases of 20.73% and 17.10% compared with WG608, respectively (Figure 5D). These results proved that an enhanced glyoxylate pathway can effectively produce L-lysine and ε-PL synthesis. In addition, studies have shown that diaminopimelic dehydrogenase (encoded by *ddh*) can shorten the synthesis pathway from aspartate to L-lysine, thereby promoting L-lysine synthesis. Liu et al. enhanced the carbon flux towards the L-lysine biosynthesis pathway by deleting *dapD* (encoding tetrahydrodipicolinate succinylase) and overexpressing *ddh* [37]. This manipulation weakened the succinylation pathway and strengthened the meso-diaminopimelate dehydrogenase pathway, ultimately increasing L-lysine production from approximately 150 g/L to 189 ± 8.7 g/L. Conversely, deleting the *ddh* and *lysE* genes significantly reduced L-lysine production while simultaneously enhancing the synthesis of its competing pathways’ products, L-threonine and L-isoleucine [48]. Therefore, exogenous *ddh^bs^* and *ddh^cg^* from *Bacillus sphaericus* and *C. glutamicum*, respectively, were amplified to assess their effects on ε-PL production. The recombinant plasmids pIB139-*ddh^bs^* and pIB139-*ddh^cg^* were subsequently constructed and transferred into WG608, and the recombinant strains WME8 and WME9 were obtained. As a result, the ε-PL production rates of WME8 and WME9 reached 2.16 ± 0.13 g/L and 2.20 ± 0.14 g/L, showing increases of 11.92% and 13.99% compared with WG608 (Figure 5C,D). These results show that introducing exogenous diaminopimelic dehydrogenase is favorable for ε-PL biosynthesis, and *ddh^cg^* from *C. glutamicum* is more beneficial to ε-PL biosynthesis than *ddh^bs^* from *B. sphaericus*.

#### 3.4.3. Constructing a High-Producing ε-PL Engineering Strain via Metabolic Engineering

Based on our integrated analysis of transcriptomes and metabolomes, metabolic engineering was used to simultaneously enhance L-lysine supply, ATP supply, and the expression of ε-PL synthase to increase the ε-PL production of WG608. The increase in L-lysine supply was achieved by enhancing the glyoxylate pathway and diaminopimelic dehydrogenase, while the increase in ATP supply was achieved by constructing an ATP recycling system. Finally, the recombinant strain WME10 was constructed after the synergistic overexpression of the above six effective genes (*aceA*, *sdhA*, *ddh^cg^*, *pap*, *ppk2B^cg^*, and *pls*) in WG608 (Figure 6A). The ε-PL production rate of WME10 reached 2.51 ± 0.11 g/L, 30.05% higher than the original strain, WG608 (Figure 6B). After 192 h of fed-batch fermentation in a 5 L bioreactor, the ε-PL production rate of WME10 reached 77.16 g/L, 74.05% higher than that of WG608 (Figure 6C,D). The intracellular ATP concentration of WME10 also significantly increased during the whole fermentation process (Figure 6E).

The aim of this study is to enhance the ε-PL production of the high-producing mutant WG608 through metabolic engineering. However, due to the unclear intrinsic mechanisms of high ε-PL production, it is challenging to identify the key pathways and genetic elements influencing ε-PL production. Therefore, comparative transcriptome and metabolome analyses of the high-yield and low-yield strains were conducted at different fermentation phases (48, 96, and 144 h) to identify the key pathways or genetic elements affecting ε-PL synthesis. Subsequently, metabolic engineering was utilized to further enhance ε-PL production. This multi-omics integrative analysis approach provides valuable references and insights for the study of ε-PL and other secondary metabolites.

Multi-omics analysis indicates that the enhanced glycolysis pathway, weakened TCA cycle, and upregulated glyoxylate cycle promote ε-PL biosynthesis in WG608. Previous studies have shown that upregulating the glyoxylate pathway increases the synthesis of metabolites in the aspartate pathway in *Escherichia coli* [49]. Van et al. also found that by reducing the flux through the TCA cycle, the metabolic flow could be redirected toward L-lysine biosynthesis, with the glyoxylate cycle playing a crucial role in this metabolic adjustment [50]. These results are similar to our results. Additionally, studies have shown that the glyoxylate cycle, as an alternative to the TCA cycle, plays a vital role in the supply of oxaloacetate and the biosynthesis of L-lysine in *C. glutamicum* [51,52]. However, the effect of the glyoxylate cycle on ε-PL synthesis was not reported. This study validated the crucial role of the glyoxylate cycle in ε-PL biosynthesis and extended its application in the synthesis of ε-PL and other secondary metabolites in *S. albulus*. Furthermore, multi-omics analysis also showed that the L-lysine synthesis pathway was downregulated in the late phase of fermentation. To further enhance L-lysine synthesis, diaminopimelate dehydrogenase from *C. glutamicum* was introduced into *S. albulus* for the first time, shortening the L-lysine biosynthesis pathway and increasing the L-lysine supply and ε-PL production. This strategy demonstrates the potential of optimizing L-lysine and ε-PL biosynthesis pathways by introducing exogenous genes, providing new insights for the construction of high-efficient microbial cell factories for ε-PL production.

Maintaining a high intracellular concentration of ATP is crucial in the process of ε-PL synthesis due to the allosteric regulation of Pls by ATP. At low concentrations (0.25–2 mM), ATP acts as a negative allosteric regulator of Pls; whereas, at high concentrations (3–5 mM), ATP serves as a positive allosteric regulator. This suggests that only high concentrations of ATP (above 3 mM) can fully activate Pls [35]. The allosteric regulation of ATP in Pls is conducive to preventing its excessive consumption in primary metabolism, thus allowing it to accumulate as an important cofactor for ε-PL biosynthesis. Due to insufficient ATP supply in the late fermentation phase of WG608, the ATP supply was enhanced by overexpressing *vgb* and constructing an ATP regeneration system. The results showed that both methods significantly increased ε-PL production. Xu et al. improved O_2_ supply by overexpressing *vgb*, thereby promoting ATP synthesis and enhancing ε-PL production, which is consistent with our results [33]. However, our study found that constructing an ATP regeneration system was more beneficial for ε-PL synthesis compared to overexpressing *vgb*, which might be because of the excessive consumption of intracellular NADH in WG608 during the late fermentation phase. Ultimately, the ATP supply was significantly increased throughout the fermentation process by constructing an ATP regeneration system, which is another important reason for the significantly increased ε-PL production in engineered strain WME10.

In summary, this study presents a multi-omics-guided metabolic engineering method to further improve ε-PL production in the high-producing *S. albulus* WG608 strain. First, an integrative analysis of transcriptomes and metabolomes was used to evaluate transcriptional and metabolic changes in high- and low-producing strains, thus identifying the key pathways and genes involved in ε-PL biosynthesis (Figure 1 and Figure 2). This result showed that upregulating the glycolysis pathway, the glyoxylate cycle, the L-lysine biosynthesis pathway, and the high expression of Pls were the main reasons for the increased ε-PL production (Figure 3). Subsequently, the key gene elements that can enhance the L-lysine synthesis pathway, increase ATP supply, and strengthen the ε-PL biosynthesis pathway were identified through overexpression (Figure 4 and Figure 5). Finally, strain WME10 was obtained through the synergistic overexpression of *ppk2B^cg^*, *pap*, *pls*, *aceA*, *sdhA*, and *ddh^cg^*. In a 5 L bioreactor, WME10 achieved an ε-PL production rate of 77.16 ± 1.03 g/L, 74.05% higher than that of the parent strain, WG608. This result represents the highest reported ε-PL production rate in *S. albulus* (Figure 6). The engineered ε-PL-producing *S. albulus* developed in this work lays the foundation for further enhancing ε-PL production via microbial fermentation, which can be used to isolate and purify ε-PL as a natural antifungal agent. As a future research direction, agricultural byproducts (such as molasses) can be used as cost-effective media to further improve production efficiency and reduce the costs of industrial ε-PL production.

## Figures and Tables

**Figure 1 biomolecules-14-00752-f001:**
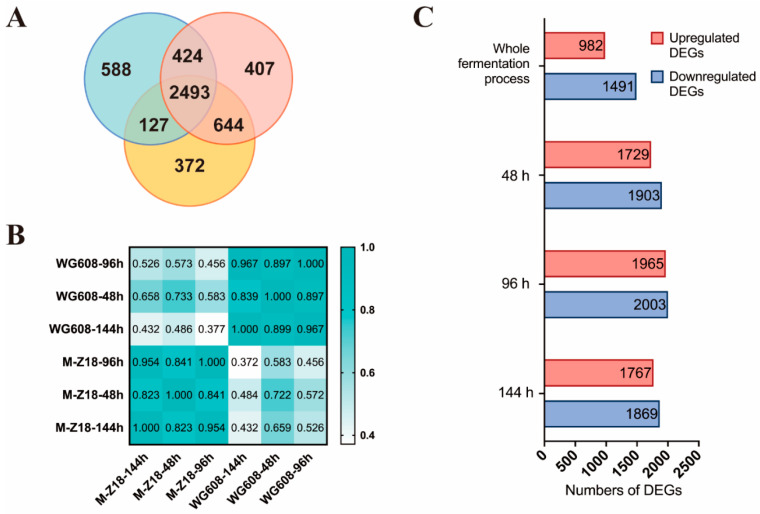
Overview of transcriptome results at different fermentation phases. (**A**) Venn diagrams of differentially expressed genes. (**B**) Pearson correlation heat map of gene expression amount. (**C**) Numbers of differentially expressed genes.

**Figure 2 biomolecules-14-00752-f002:**
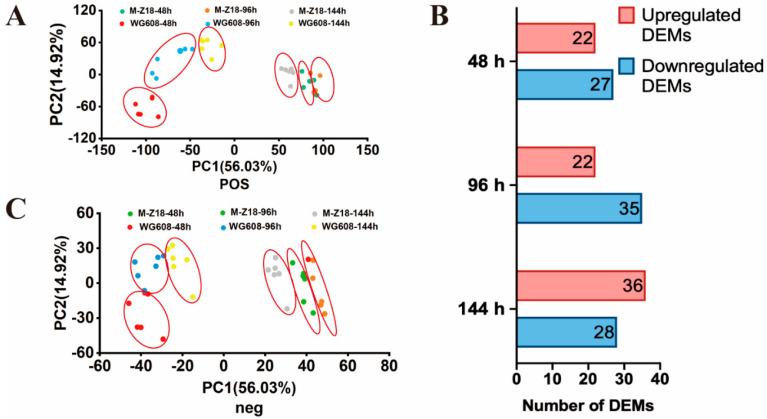
Overview of metabolome results at different fermentation phases. (**A**) Principal component analysis of the metabolome in positive ion mode. (**B**) Principal component analysis of the metabolome in negative ion mode. (**C**) Numbers of differential expressed metabolites.

**Figure 3 biomolecules-14-00752-f003:**
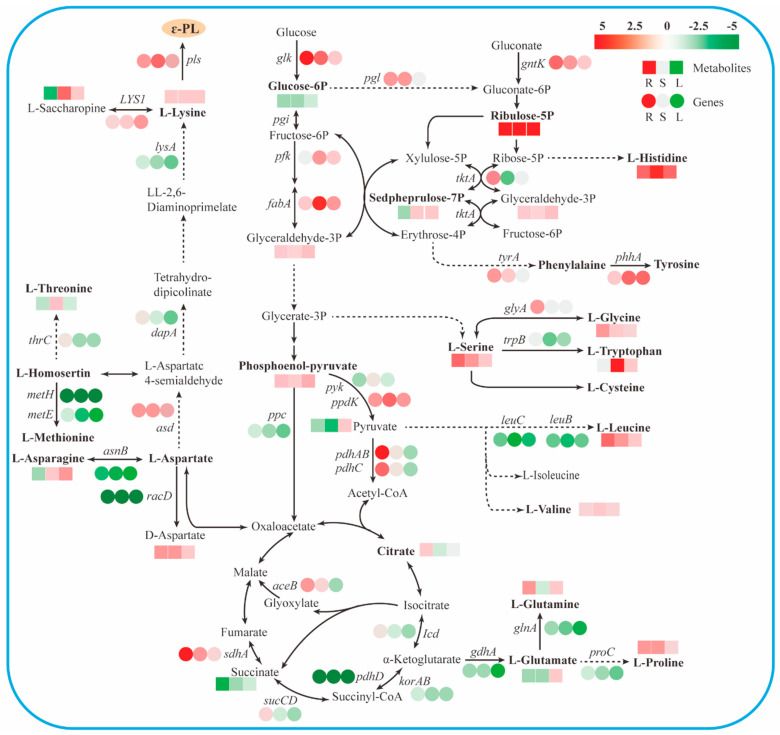
Transcriptional and metabolic changes in the ε-PL biosynthesis pathway. Squares represent the metabolome; circles represent the transcriptome; red represents upregulated expression of genes/increased accumulation of metabolites; green represents downregulated expression of genes/decreased accumulation of metabolites.

**Figure 5 biomolecules-14-00752-f005:**
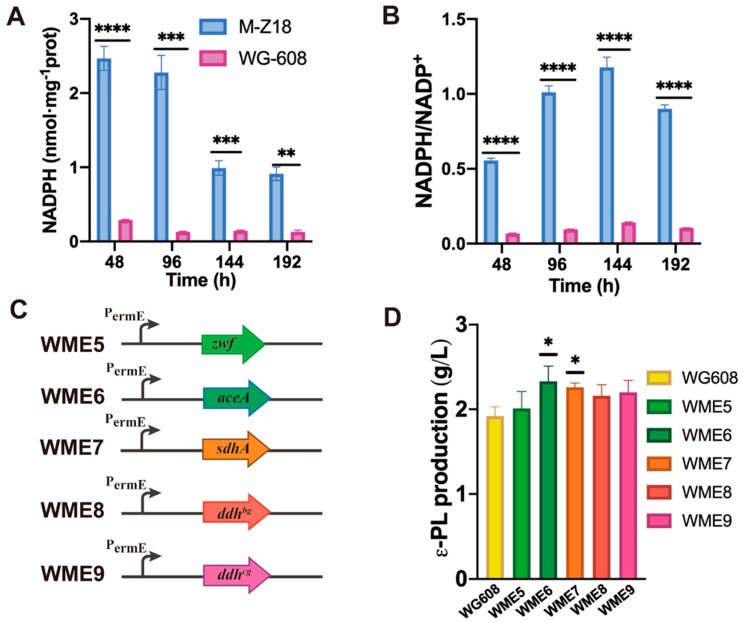
Effect of the overexpression of *zwf*, *aceA*, and *sdhA* on ε-PL production. (**A**) NADPH concentration of strains WG608 and M-Z18 during fermentation. (**B**) NADPH/NADP^+^ ratio of strains WG608 and M-Z18 during fermentation. (**C**) Schematic of the construction of strains WME5-9. (**D**) Effect of the overexpressing different genes on ε-PL production. *aceA*, *sdhA*, *ddh^bg^*, and *ddh^cg^* represent the genes encoding isocitrate lyase, succinate dehydrogenase flavoprotein subunit, diaminopimelic dehydrogenase from *Bacillus sphaericus*, and diaminopimelic dehydrogenase from *Corynebacterium glutamicum*. All data were from biological triplicates. Error bars represent standard deviation. The statistical analysis was performed by one-way ANOVA analysis; *, **, ***, and **** indicate *p* ≤ 0.05, *p* ≤ 0.01, *p* ≤ 0.001, and *p* ≤ 0.0001 relative to the control (WG608), respectively.

**Figure 6 biomolecules-14-00752-f006:**
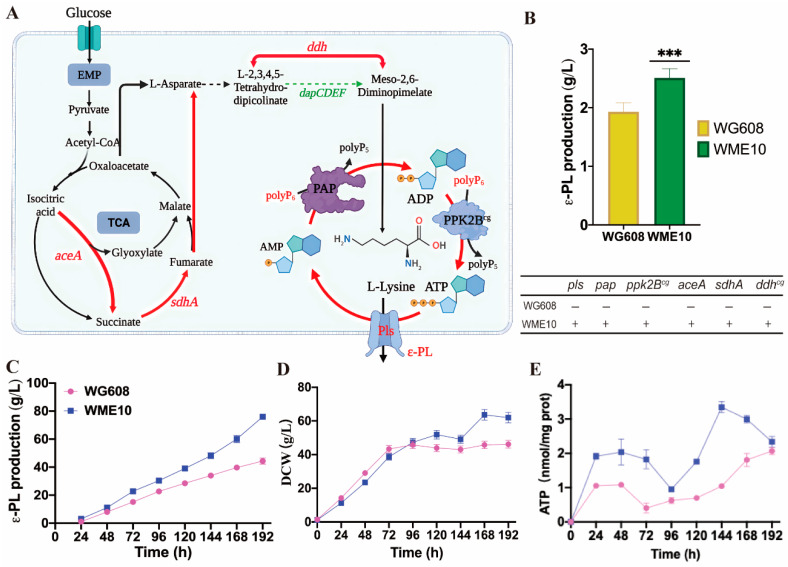
Construction of engineering WME10 and its fermentation performance. (**A**) ε-PL biosynthesis pathway in *S. albulus,* red lines represent genes overexpressed in WME10. (**B**) Shake-flask fermentation performance of WME10 and WG608. (**C**–**E**) ε-PL production, DCW, and ATP concentration of WME10 and WG608 in the fed-batch fermentation. ε-PL, ε-poly-L-lysine; DCW, cell dry weight; ATP, adenosine triphosphate. The red dots represent WG608, and the purple square represents WME10. The data are presented as averages, and the error bars represent standard deviations (n = 3 independent experiments). The statistical analysis was performed by one-way ANOVA analysis; *** indicates *p* ≤ 0.001 relative to the control (WG608), respectively.

**Table 1 biomolecules-14-00752-t001:** Strains and plasmids used in this study.

Strains and Plasmids	Description
Strains	
*S. albulus* WG608	Original strain, ε-poly-L-lysine producer
*S. albulus* WME1	WG608 carrying pIB139-*vgb*
*S. albulus* WME2	WG608 carrying pIB139-*ppk2B^cg^-pap*
*S. albulus* WME3	WG608 carrying pIB139-*pls*
*S. albulus* WME4	WG608 carrying pIB139-*ppk2B^cg^-pap-pls*
*S. albulus* WME5	WG608 carrying pIB139-*zwf*
*S. albulus* WME6	WG608 carrying pIB139-*aceA*
*S. albulus* WME7	WG608 carrying pIB139-*sdhA*
*S. albulus* WME8	WG608 carrying pIB139-*ddh^bs^*
*S. albulus* WME9	WG608 carrying pIB139-*ddh^cg^*
*S. albulus* WME10	WG608 carrying pIB139-*ppk2B^cg^-pap-pls-aceA-sdhA-ddh^cg^*
*E.coli* DH5α	General cloning host
*E.coli* ET12567	Donor strain for conjugation between *E.coli* and *Streptomyces*
Plasmids	
pIB139	Integrative vector based on *ϕ*C31 integrase
pIB139-*vgb*	*vgb* cloned in pIB139
pIB139-*ppk2B^cg^-pap*	*ppk2B^cg^* cloned in pIB139
pIB139-*pls*	*pap* and *ppk2B^cg^* cloned in pIB139
pIB139-*ppk2B^cg^-pap-pls*	*pls, pap,* and *ppk2B^cg^* cloned in pIB139
pIB139-*zwf*	*zwf* cloned in pIB139
pIB139-*aceA*	*aceA* cloned in pIB139
pIB139-*sdhA*	*sdhA* cloned in pIB139
pIB139-*ddh^bs^*	*ddh^bs^* cloned in pIB139
pIB139-*ddh^cg^*	*ddh^cg^* cloned in pIB139
pIB139-*ppk2B^cg^-pap-pls-aceA-sdhA-ddh^cg^*	*aceA*, *sdhA*, *ddh^cg^*, *pap*, *ppk2B^cg^,* and *pls* cloned in pIB139

**Table 2 biomolecules-14-00752-t002:** Differential expression genes between *S. albulus* WG608 and *S. albulus* M-Z18.

Pathway	Gene Name	Description	Gene ID	Fold Change (log_2_)		
				48 h	96 h	144 h
EMP	*glk*	glucokinase	M-Z18AGL006854	2.76	2.10	1.48
	*pfk*	6-phosphofructokinase	M-Z18AGL006038	−0.10	1.60	1.39
	*fbaA*	fructose-bisphosphate aldolase, class II	M-Z18AGL001662	0.56	3.62	2.13
	*tpiA*	triosephosphate isomerase (TIM)	M-Z18AGL001535	4.08	−1.81	−0.34
PPP	*pgl*	6-phosphogluconolactonase	M-Z18AGL001375	1.28	1.13	−0.005
	*gntK*	gluconokinase	M-Z18AGL003471	2.89	2.63	1.63
	*tktA*	transketolase	M-Z18AGL008451	2.53	−2.23	−0.10
TCA	*pyk*	pyruvate kinase	M-Z18AGL006160	−1.24	0.43	−0.55
	*ppdK*	pyruvate, orthophosphate dikinase	M-Z18AGL005661	2.27	2.79	2.40
	*ppc*	phosphoenolpyruvate carboxylase	M-Z18AGL005090	−0.95	−1.23	−1.68
	*aspB*	aspartate aminotransferase	M-Z18AGL001941	1.64	1.13	0.62
	*mdh*	malate dehydrogenase	M-Z18AGL003522	−1.02	−1.03	−1.61
	*acs*	acetyl-CoA synthetase	M-Z18AGL005013	3.34	3.27	3.99
	*icd*	isocitrate dehydrogenase	M-Z18AGL006751	0.27	−0.26	−1.10
	*korA*	2-oxoglutarate/2-oxoacid ferredoxin oxidoreductase subunit alpha	M-Z18AGL004791	−0.73	−1.41	−1.19
	*korB*	2-oxoglutarate/2-oxoacid ferredoxin oxidoreductase subunit beta	M-Z18AGL004790	−0.53	−1.00	−1.01
	*pdhD*	dihydrolipoyl dehydrogenase	M-Z18AGL007849	−9.21	−9.75	−9.59
	*purA*	adenylosuccinate synthase	M-Z18AGL004630	1.40	1.64	1.15
	*sucC*	succinyl-CoA synthetase beta subunit	M-Z18AGL003534	0.75	−0.70	−1.01
	*sdhA*	succinate dehydrogenase flavoprotein subunit	M-Z18AGL006117	5.33	1.98	1.17
	*aceA*	isocitrate lyase	M-Z18AGL007682	1.45	1.72	−1.45
	*aceB*	malate synthase	M-Z18AGL007679	1.59	1.05	−1.41
DAP	*aspB*	aspartate aminotransferase	M-Z18AGL001941	1.64	1.13	0.62
	*racD*	aspartate racemase	M-Z18AGL007833	−13.33	−11.08	−14.57
	*asd*	aspartate-semialdehyde dehydrogenase	M-Z18AGL005529	1.38	1.44	1.03
	*dapA*	4-hydroxy-tetrahydrodipicolinate synthase	M-Z18AGL005888	0.05	−0.44	−1.02
	*lysA*	diaminopimelate decarboxylase	M-Z18AGL004313	−0.51	−0.80	−1.53
	*lysS*	lysyl-tRNA synthetase	M-Z18AGL001476	0.32	1.30	1.67
ε-PL biosynthesis pathway	*pls*	ε-PL synthetase	M-Z18AGL001343	1.89	2.47	1.13

## Data Availability

All data generated or analyzed during the current study are included in this manuscript and Appendix A.

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
