# Peer review of "Enhanced ε-Poly-L-Lysine Production in Streptomyces albulus through Multi-Omics-Guided Metabolic Engineering"

_biomolecules, 2024, doi:10.3390/biom14070752_

Round 1

Reviewer 1 Report

Comments and Suggestions for Authors

The content of this manuscript entitled as “High-Level ε-Poly-L-Lysine Production in Streptomyces albulus by Multi-Omics-Guided Metabolic Engineering” seems appropriate for this journal.

  • Authors should recheck the title it is not an attractive topic if possible please revise it to attract the reader's attention.
  • Please revise the first line of the abstract. Start it with the aim or importance of the study and if you want to use this line it could be written as “Foodborne diseases caused by pathogens result in hundreds and thousands of deaths annually” or better to write the value here.
  • Again the second line is grammatically not correct there use of inappropriate verbs in this sentence.
  • In line 13 authors should avoid using subjective nouns like “we” etc.
  • Please add a statistical model and analysis that has been used for the interpretation of results.
  • Add a proper conclusive line at the end of the abstract that should summarize the abstract.

·         The sequence of introduction needs some improvements firstly mention the importance of poly-L-Lysine then correlate it with food-borne diseases and its other applications.

·         In line 71 what do you mean by “In previous study, a high-producing strain,”

·         Please add some studies that explains about the safety of É›-PL, and also add some studies that should mention its applications in the food and health industry.

·         Add a rationale and reasoning of the study in the last paragraph of the introduction section.

·         There are a lot of lines that have no sequence and not clear to understand like.

·         Please add references in all the methods that have been adopted to exhibit this study.

·         Please check the whole article avoid to use subjective nouns like “we” which has been repeatedly used.

·         In Table 1. Strains and plasmids used in this study last column sources and references please remove the word this study and replace it with other suitable word or no need to write anything.

·         In line 206 “Each 5 mL of sample was divided into two equal parts” it could be written as 5 mL of each sample was divided inti two equal: please improve its grammar that really needs serious attentions.

·         In line 216 “Ultra-Performance Liquid Chromatography (UPLC) analysis was conducted at 45°C” was should be replaced with “were”.

·         No references in metabolomic analysis, also mention the complete details of UPLC model.

·         The results section is well explained but the discussion needs a lot of effort, there should be proper correlation and reasoning along with the justification and comparison.

overall plagiarism should also be removed 36% is not acceptable according to rules.

Comments on the Quality of English Language

English is very difficult to understand/incomprehensible

Author Response

Q1: Authors should recheck the title it is not an attractive topic if possible please revise it to attract the reader’s attention.

A: Thanks for the reviewer’s kind reminder. The title has been changed to “Enhanced ε-Poly-L-Lysine Production in Streptomyces albulus through Multi-Omics-Guided Metabolic Engineering”.

Q2: Please revise the first line of the abstract. Start it with the aim or importance of the study and if you want to use this line it could be written as “Foodborne diseases caused by pathogens result in hundreds and thousands of deaths annually” or better to write the value here.

A: Thanks for your kind advice. The first line of the abstract was revised as “Safe and eco-friendly preservatives are crucial to preventing food spoilage and illnesses, as food-borne diseases caused by pathogens result in approximately 600 million cases of illness and 420,000 deaths annually”. (P1L9-11)

Q3: Again the second line is grammatically not correct there use of inappropriate verbs in this sentence.

A: Thank you for your feedback. We have revised the second sentence and combined it with the first sentence for improved clarity and conciseness. (P1L9-11)

Q4: In line 13 authors should avoid using subjective nouns like “we” etc.

A: We keenly accept the suggestion and rewrite this sentence as below: ε-PL’s biosynthetic capacity was enhanced in Streptomyces albulus WG608 through metabolic engineering guided by multi-omics techniques. (P1L13-14)

Q5: Please add a statistical model and analysis that has been used for the interpretation of results.

A: Thank you. The statistical model and analysis that has been used for the interpretation of results have been present in P6L256-263.

Q6: Add a proper conclusive line in the end of abstract that should summarize the abstract.

A: Thanks for your kind advice. We have added the conclusive line in the end of abstract as below: These results suggest that the integrative analysis of the transcriptome and metabolome can facilitate the identification of key pathways and genetic elements affecting ε-PL synthesis, guiding further metabolic engineering, thus significantly enhancing ε-PL production. (P1L20-23)

Q7: The sequence of introduction needs some improvements firstly mention the importance of poly-L-Lysine than compare it with food-borne diseases and its other applications.

A: We have changed the sequence of the introduction, and the revised introduction is as follows:

ε-Poly-L-lysine (ε-PL) is a bioactive molecule composed of 25-35 L-lysine residues linked through α-carboxyl and ε-amino groups. It is known for its broad-spectrum an-timicrobial activity, water solubility, biodegradability, and high safety. Studies have shown that ε-PL is non-toxic in the human body and can be fully metabolized (Tan et al., 2019). Additionally, it is stable at high temperatures and does not form harmful by-products (Xu et al., 2019). These excellent properties have led to its widespread use as a food preservative in several countries (Hamano et al., 2011). The significance of ε-PL as a food preservative lies in its ability to effectively combat the growth of foodborne patho-genic bacteria, which cause approximately 600 million illnesses and 420,000 deaths annually (Crowe-Mcauliffe et al., 2021; Yeon et al., 2022). For example, ε-PL can effec-tively inhibit the growth of Escherichia coli O157 and Listeria monocytogenes, thereby significantly improving food safety and reducing the risk of foodborne diseases (Shi et al., 2016). Besides inhibiting the growth of various microorganisms, ε-PL also maintains the sensory quality and nutritional value of foods. For instance, adding ε-PL to fresh juices can significantly extend their shelf life and prevent microbial contamination (Wang and Rong, 2022). In addition to its extensive applications in the food industry, ε-PL also shows great potential in the medical field. For example, ε-PL can be used to create hydrogels with self-healing and antibacterial properties, indicating its broad applications in wound repair and tissue engineering (Wang et al., 2017). Research also indicates that ε-PL can bind to G-quadruplex DNA structures and effectively mediate targeted drug and gene delivery. These properties make ε-PL highly promising for anticancer drug delivery and gene therapy (Marzano et al., 2020). (P1L28-49)

Q8: In line 71 what do you mean by “In previous study, a high-producing strain,”

A: This sentence means: In our previous study, a high-producing strain, S. albulus WG608, was obtained through successive rounds of mutagenesis and ribosome engineering. We have revised this sentence and cited our previously published literature to support it (Wu et al., 2016) (P2L85-86).

Q9: Please add some studies that explains about the safety of É›-PL, and also add some studies that should mention its applications in the food and health industry.

A: Thanks for your kindly advice. The introduction have been revised as requested and marked in red. (P1L28-49)

Q10: Add a rationale and reasoning of the study in the last paragraph of the introduction section.

A: Thanks for your advice. we have added a rationale and reasoning of the study in the last paragraph of the introduction section. (P2L85-99)

Q11: There are a lot of lines that have no sequence and not clear to understand like.

A: Thank you for your suggestions. We have carefully revised the manuscript to make it more logical. Additionally, we had the language professionally edited to make the paper more coherent and clearer.

Q12: Please add references in all the methods that have been adopted to exhibit this study.

A: Thanks for your kindly advice. We have added references in all the methods that have been adopted to exhibit this study, and marked them in red.

Q13: Please check the whole article avoid to use subjective nouns like “we” which has been repeatedly used.

A: Thanks for your advice. we have checked the whole article to avoid using subjective nouns like “we”.

Q14: In Table 1. Strains and plasmids used in this study last column sources and references please remove the word this study and replace it with other suitable word or no need to write anything.

A: Thank you. Table 1 has been changed according to the suggestion.

Q15: In line 206 “Each 5 mL of sample was divided into two equal parts” it could be written as 5 mL of each sample was divided inti two equal: please improve its grammar that really needs serious attentions.

A: Thanks for your kindly advice. “Each 5 mL of sample was divided into two equal parts” it could be written as 5 mL of each sample was divided inti two equal. (P6L222-223)

We carefully checked the article multiple times and had professionals polish the language to avoid any grammatical issues.

Q16: In line 216 “Ultra-Performance Liquid Chromatography (UPLC) analysis was conducted at 45°C” was should be replaced with “were”.

A: Thanks for your advice. We have revised this paragraph.(P6L232-243)

Q17: No references in metabolomic analysis, also mention the complete details of UPLC model.

A: Thanks for your kindly advice. We have added some supplementary references in metabolomic analysis, and the details of UPLC model was demonstrated in section 2.6. The modifications are marked in red. (P6L232-243).

Q18: Results section is well explained but discussion needs a lot of efforts, there should be proper correlation and reasoning along with the justification and comparison.

A: We have improved the discussion section of the manuscript according to the reviewer’s suggestion, and the modifications are marked in red.

Q19: Overall plagiarism should also be removed 36% is not acceptable according to rules.

A: Thank you for your suggestion. We have carefully revised the manuscript and reduced the self-citation rate as much as possible.

Reviewer 2 Report

Comments and Suggestions for Authors

The manuscript submitted for review concerns the preparation of overexpression plasmids of a selected enzyme in a bacterial overexpression system.

The aim of the work is clear, the research was planned and carried out according to the art of conducting research of this type, the techniques used are not objectionable.

However, the paper raises a number of issues for clarification and improvement:

1. Figure 3 is the metabolic pathway proposed by the authors of the study or based on the literature? It is worth indicating whether this is a study by the authors or based on whom?

2. The number of self quotations exceeded! I realise that this is a continuation of other studies but please verify the number of self-citations.

3. Why did the authors opt for the tent type of overexpression. A simpler and more stable system based on E. coli or Pichia yeast could have been chosen?

Author Response

Q1. Figure 3 is the metabolic pathway proposed by the authors of the study or based on the literature? It is worth indicating whether this is a study by the authors or based on whom?

A:Thanks for your advice. We are sorry for not describing it clearly.

The metabolic pathway in Figure 3 was constructed in this study based on the genomic data of Streptomyces albulus ZPM that has been reported in the NCBI (GenBank: CP006871.1). Based on the pathway, we analyzed the transcriptional and metabolic changes in high-producing strain WG608 compared to the control. We also made some modifications in the manuscript to make it clearer. The modified section are marked in red (P9L324-327).

Q2. The number of self-quotations exceeded! I realise that this is a continuation of other studies but please verify the number of self-citations.

A: Thank you for your suggestion. We have carefully revised the manuscript and reduced the self-citation rate as much as possible.

Q3. Why did the authors opt for the tent type of overexpression. A simpler and more stable system based on E. coli or Pichia yeast could have been chosen?

A: Thanks for your advice. Sorry for our unclear description on this point. The tent type of overexpression (conjugal transfer) is specifically used to the metabolic and regulatory environment of Streptomyces albulus, which is the host strain in our study. Using a system based on E. coli or Pichia yeast might not replicate the unique intracellular conditions necessary for optimal ε-PL production. Therefore, conjugal transfer was used for overexpression to further improve the ε-PL production of the high-producing strain.

Reviewer 3 Report

Comments and Suggestions for Authors

The manuscript titled with " High-Level ε-Poly-L-Lysine Production in Streptomyces albulus by Multi-Omics-Guided Metabolic Engineering ". The manuscript discusses a good point. Overall, the presented study indicates for enhance ε-PL synthesis in Streptomyces albulus through metabolic engineering guided by multi-omics techniques. These findings present innovative methods to increase ε-PL production that could be applicable to other valuable natural antibacterial agents. The manuscript written well. But it needs a minor revision as follows: 

1- Abstract:

The code number for the bacterial strains used in the Abstract must be included

2- Introduction

The research objective must be clarified at the end of the introduction and also in the conclusion

3- Materials and Methods

The authors should use cheap fermentation media and the cost of lysine production is economical and inexpensive Such as molasses and other.

Author Response

Q1- Abstract:

The code number for the bacterial strains used in the Abstract must be included

A: Thanks for your kindly advice. We have added the code number for the bacterial strains used in the Abstract. (P1L13)

2- Introduction

The research objective must be clarified at the end of the introduction and also in the conclusion

A: Thank you for your suggestions, which have been very helpful for improving our article. We have modified the introduction and the conclusion, and clarifying the purpose of this study. The modifications are marked in red. (P2L85-99; P18L584-601)

3- Materials and Methods

The authors should use cheap fermentation media and the cost of lysine production is economical and inexpensive Such as molasses and other.

A: Thank you for your valuable suggestion. We strongly agree with the importance of using cheap fermentation media, such as molasses and other agricultural by-products. However, due to current experimental conditions and time constraints, we are unable to conduct molasses fermentation experiments at this time. Nevertheless, we plan to further pursue this approach in future research. We will strive to explore and optimize the fermentation conditions using molasses and other agricultural by-products as cost-effective media to improve production efficiency and reduce costs. We have also added this future research direction to the conclusion section. (P16L599-601)

Round 2

Reviewer 1 Report

Comments and Suggestions for Authors

Abstract is an important part of the article, Most of the readers get idea of the article by only reading the abstract. It needs some more efforts. Please add a single line which summarize the statistical analysis of your study. 

Some latest study studies should be added in the introduction section as there is no reference from the current year.

in discussion section justification and reasoning is missing, please recheck and give at least 1 reason of enhancement or improvement in your results.

Comments on the Quality of English Language

Moderate editing of English language required

Author Response

Q1. Abstract is an important part of the article, Most of the readers get idea of the article by only reading the abstract. It needs some more efforts. Please add a single line which summarize the statistical analysis of your study. 

A: Thanks for your kindly advice. We have add the statistical analysis of our study in the abstract as below:

Based on transcriptome and metabolome data, differentially expressed genes (Fold Change >2 or <0.5; p < 0.05) and differentially expressed metabolites (Fold Change >1.2 or <0.8) were separately subjected to Gene Ontology (GO) and Kyoto Encyclopedia of Genes and Genomes (KEGG) pathway enrichment analysis. (P1L14-17)

Q2. Some latest study studies should be added in the introduction section as there is no reference from the current year.

A: Thank you very much for your valuable suggestion. We have revised the introduction section and have incorporated references from current year. The modifications are marked in red. (P1L34-42)

Q3. In discussion section justification and reasoning is missing, please recheck and give at least 1 reason of enhancement or improvement in your results.

A: Thanks for your kindly advice. We have rechecked the discussion section, and give several reasons of enhancement in our results. The modifications are marked in red. (P17L565-610)